# Exploring the Evacuation of Dairy Cattle at Night in Collaboration with the Fire Brigade: How to Prepare Openings for Swift Rescue in Case of Barn Fire

**DOI:** 10.3390/ani12111344

**Published:** 2022-05-25

**Authors:** Florian Diel, Elke Rauch, Rupert Palme, Carola Sauter-Louis, Eva Zeiler

**Affiliations:** 1Faculty of Sustainable Agricultural- and Energy Systems, University of Applied Sciences Weihenstephan-Triesdorf, 85354 Freising, Germany; eva.zeiler@hswt.de; 2Chair of Animal Welfare, Ethology, Animal Hygiene and Animal Husbandry, Department of Veterinary Sciences, Faculty of Veterinary Medicine, LMU Munich, 80539 Munich, Germany; rauch@lmu.de; 3Department of Biomedical Sciences, University of Veterinary Medicine, 1210 Vienna, Austria; rupert.palme@vetmeduni.ac.at; 4Institute of Epidemiology, Friedrich-Loeffler-Institut, 17493 Greifswald, Germany; carola.sauter-louis@fli.de

**Keywords:** livestock evacuation, fire preparedness, stockmanship, husbandry, sensory physiology, emergency management

## Abstract

**Simple Summary:**

The evacuation of farm animals out of a barn is a rarely considered subject. Especially in case of fire, there is a need for functional emergency exits as well as concepts of rescue for swift evacuation, since both the harmful smoke and the danger of collapsing roofs call for urgency. Field reports of firefighters and affected farmers state that barn animals hesitate to leave their familiar surroundings and rather try to withdraw to their known housing, which they deem as safe. Thus, it is not sufficient to simply open the doors and gates hoping for self-rescue of the animals. As there is a lack of guidelines on the design of emergency exits, we conducted an evacuation exercise of year-round housed dairy cattle, in cooperation with the fire brigade by night, to inspect the animals’ behaviour. We found that preparing the exits according to the sensory perception of cattle, providing familiar surfaces and adequate lighting, and herding cattle correctly result in a speedy rescue, even if the animals were not habituated to leaving the barn before. We hope to set an example for farmers and fire brigades that are in need of instructions, and that more research considering this vital topic will follow.

**Abstract:**

The aim of this study was to investigate the influencing factors of successfully rescuing year-round housed cattle in case of a barn fire. Empirical research indicates the reluctance of cattle to leave their familiar barn. Subsequent retreat back to the perceived safety inside, which stands in contrast to the unknown and thus adversary elements outside, for example, the fire brigade, is to be expected. We examined the evacuation of 69 dairy cattle, split into three groups, to an adjacent pasture by night and inspected the animals’ acceptance of two differently designed escape routes and the effect of preceding training. Along with the time needed for evacuating all animals, we measured faecal cortisol metabolites and daily milk yield to assess stress in the animals. Our preliminary assumption was that cattle trained for pasture would have a decisive advantage over untrained cattle. However, adapting the exits to the sensory physiology of the cattle resulted in an extensive impact on the animals’ readiness to leave the familiar housing, as the evacuation of the cattle non-habituated to the exit was comparatively quick and successful. We consider this study instructional for fire brigades and farmers, encouraging them to develop a customised concept for rescuing their cattle in case of an emergency.

## 1. Introduction

The research problem, concerning the evacuation of cattle in case of a barn fire, is an apparent gap in the current state of knowledge. Statistics about the occurrence of barn fires are scarce. While some data are collected about disasters by public authorities, the focus is usually on human fatalities or economic damages expressed by the value lost [1,2,3]. Concerning barn fires, the data collected vary vastly between countries, forming a patchwork of less expressive statistics. The number of fires in the agricultural sector in Austria is second in ranking right after residential fires [4]. In The Netherlands, between 0.2% and 0.3% of cattle farms, 0.5% of pig farms, and up to 1.28% of poultry farms are affected by barn fires per year [5]. In Belgium, around 4% of all fires were barn fires [6]. In Germany, the umbrella association of insurance companies counts about 5000 barn fires per year, being around 2.5% of all fires recorded, while there is no further differentiation between species and the number of affected animals [7].

There are many media reports concerning barn fires that are reviewed by animal welfare organisations [8,9], which in turn demand revised construction codes for barns, such as obligatory sprinkler systems or fire detection systems. While guidelines exist for the prevention of barn fires [10], there are no instructions on how to best evacuate livestock or how to design exits. The World Organisation for Animal Health (OIE) only mentions the general need for evacuation plans in their Terrestrial Animal Health Code [11] (Article 7.11.17 No. 16). Further recommendations limit themselves to specifying the minimum width of exits to be 1.5–2 times the largest animal width or elucidating the amount of exits and the maximum travel distance to an exit, with no further instructions given [12,13].

Farmers who are at risk of suffering from bush fires are given information about the logistics of transporting groups of animals through the countryside [14,15], triage references for assessing cattle after a fire (i.e., when fences trap the animals) [16], and feeding them afterwards [17]. However, the animals’ reaction towards wildfires cannot be considered illustrative concerning the situation in a constricted space such as a barn, since the animals in an open field have plenty of options to evade the fire or even take advantage of it in hunting or foraging [18,19].

Regarding the behaviour of farm animals during a barn fire, there are only empirical field reports from affected farmers and firefighters. Factsheets underline the variety of animal behaviours, ranging from panic to aggression [20,21]. In addition, stoicism following exposure to smoke can result from early stages of carbon monoxide poisoning [22]. With respect to horses, guidance on blindfolding the animals and leading them out of the barn individually highlights the problematic instinct of the animals to seek refuge in their familiar housing [23]. The same instinct should be expected in other farm animals, but to a lesser extent if the animals are used to pasture [24,25]. In general, the willingness to accept an exit is improved by the habituation of the animals to it. Evacuation routines for livestock were proposed to become obligatory in East Germany [26]. In the 1980s, experimental evacuations of livestock took place in Russia [27]. Ruppert cites these studies in his dissertation, describing the observations that cattle habituated to pasture left the barn on their own, but only through the known openings. In contrast, non-habituated cattle took greater effort in terms of time and manpower [24] (pp. 56–58).

These field reports are congruent with prevailing opinions about herding and handling cattle, based on their physiology, ethology, and sensory perception. There are similarities between how to evacuate cattle in the case of a fire and how to reduce stress within cattle at the slaughter plant. In both cases, it is necessary to herd stressed cattle to an unknown location. Thus, the extensive work of Temple Grandin about welfare audits and handling and herding groups of cattle must be considered. Cattle choose an unpleasant but known option rather than one unknown to them [28]. Considering the sight of cattle, the adaptation to light is up to five times slower than in humans [29]. A path orderly lighted by the fire brigade at night or bright sunshine in the daytime might be glaring and blinding for cattle. In addition, the depth perception of cattle is worse than of humans, resulting in the need for cows to inspect sharp contrasts on the floor, such as those created by shadows [30]. Because of this, the movement of a group of cows might be hindered or even stopped. In order for cows to accept races, for example, at the slaughter plant, they need safe footing and a clear line of sight [31]. The same is true for unknown flooring. In general, every distraction such as deflections, unknown vehicles, or persons around the exit can result in cattle balking, refusing to move, or turning around [32]. This knowledge should also be applicable to evacuating in case of a barn fire.

To evaluate the success of an evacuation, the primary concern is the required time. Secondarily, the assessment of stress in the cattle can be an indicator of the willingness to use the escape route. However, the direct method of assessing stress by collecting blood samples right after evacuation and quantifying the serum cortisol level would not be possible, as catching and immobilising the animals on pasture for blood sampling risks additional superimposing stress [33]. Instead, quantifying faecal cortisol metabolites (FCMs) is a well-established, non-invasive approach in objectively comparing stress responses in animals [34,35,36,37]. Furthermore, due to the delay of faecal excretion, sampling faeces for quantifying FCMs must take place several hours (cattle: ~9–12 h) after the stressful event, as summarised by Palme [38].

In an effort to reduce the number of farm animals perishing in barn fires, this pilot study was undertaken. The practical goal of the study was to explore possible designs of egress and strategies evacuating cattle. The supporting, more scientific goal is to explore stress responses in cattle during evacuation. Our approach was to simultaneously evacuate three separated groups of cattle at night in cooperation with the fire brigade, advancing realistically with sirens and flashing lights, comparing two differently designed exits and the effect of habituating cattle to the exit beforehand. The long-term objective of this research topic is to establish the best practices within barns for emergencies. The aim of this study, therefore, was to indicate promising designs of openings for evacuation by recording behavioural and physiological stress responses of cattle during their rescue, thus encouraging farmers to develop rescue concepts for their farms.

## 2. Materials and Methods

### 2.1. Ethical Note

This study was approved by the competent authority of the administration of Upper Bavaria, Germany, with the internal approval code “ROB-55.2-2532. Vet_02-21-40” in compliance with the convened Ethical Committee for the use of experimental animals according to § 15 Animal Protection Law, Germany.

### 2.2. Animals and Housing

Since to the authors’ best knowledge, no comparable study exists, it was difficult to predict the behaviour of the animals used in this study during evacuation. Thus, we needed to choose the amount of stressors carefully, in accordance with the principle of refinement. For this pilot study, we focused on dairy cattle because of the advantageous conditions for rescue with this kind of husbandry. Dairy cattle are used to being handled, the housing is less subdivided, and the animal density in the stable is lower than in beef cattle or with other livestock.

The study was carried out in September 2021 at the educational and research farm Achselschwang in Bavaria, Germany. After weaning, the young cattle were raised at another site, having partial access to pasture. Right before calving, they returned to the main farm and were included in the lactating herd postpartum. They were housed year-round, with no access to pasture, in a sideways open freestall barn with deep-bedded cubicles and rubber matted flooring in the cubicle alleys. Animals were fed a total mixed ration with grass and corn silage, with additional concentrate feeding individually at feeding stations. Milking took place twice a day in a double eight-herringbone parlour.

In preparation for evacuation, one group of cows was habituated to using an exit out of the barn and up to a pasture (HABIT, *n* = 23). This was possible without repeatedly selecting them from all lactating cows, because they formed the low-performance group on the farm. Two other groups of cows were not habituated to leaving the barn. Between them, the means of egress differed, with one group exiting through a single-file race (NonH-R, *n* = 23) and the other group exiting through a wider opening (NonH, *n* = 23). Cows were assigned randomly to NonH and NonH-R shortly prior to evacuation.

Breeds in HABIT, NonH, and NonH-R were mixed with mostly Simmental (S) or Brown Swiss (B) and a few Red Holstein (RH) or hybrids with beef cattle (beef) (S-B-RH-beef; 11-6-2-4 vs. 11-9-0-3 vs. 8-8-4-3). The average age of the HABIT group differed from NonH and NonH-R (mean ± SD; 4.64 ± 2.89 vs. 5.25 ± 1.65 vs. 5.47 ± 1.23 years), as did parity (2.57 ± 2.70 vs. 3.22 ± 1.56 vs. 3.43 ± 1.06 lactations), daily milk yield (28.05 ± 3.70 vs. 41.28 ± 5.09 vs. 42.01 ± 3.55 kg), and days in milk (181 ± 95 vs. 112 ± 69 vs. 123 ± 69 days). Cows in the last trimester of gestation were not included in this study.

Initially and repeatedly during the habituation period of HABIT, cows’ health status was assessed, with defined study abort criteria. Assessment of cows in NonH-R and NonH took place in the morning of the day of evacuation. All cows were clinically healthy, had no signs of lameness, and had a BCS between 2.5 and 4 (mean ± SD; 3.08 ± 0.36 vs. 3.07 ± 0.26 vs. 2.92 ± 0.32). On the day after evacuation, lameness and general condition were assessed again, with only one cow having a contusion at an udder quarter, which was treated locally.

### 2.3. Design of Egress

The area in front of the barn was an asphalt surface of 20 m width, adjacent to around 0.8 hectares of corralled pasture. Blue and white striped barrier tape was fixed in short distances to the pasture fence for improved visibility. Cows in HABIT went on pasture after milking in the morning for eight days prior to evacuation, using the same means of egress. They remained on pasture for 45 to 60 min before returning to the barn.

For evacuation, each group had its own escape route. HABIT and NonH-R were positioned in 2.5 m wide cubicle alleys, while NonH was positioned in the 4 m wide feed passage. Crossover passages between the cubicle alleys were closed off. Between cubicles of HABIT, NonH, and NonH-R, screening walls were put up to prevent reciprocal influences between the groups. At the end of the cubicle alleys, openings in the wall of the barn were closed off by swinging gates of 2.6 m width, while a 2.8 m swinging gate closed off the feed passage. Since all gates swung only inwards, they had to be opened before the start of evacuation to prevent trapping cows. The openings were barred by lashing straps.

The single-file race for use of NonH-R was formed out of interlocking panels (Panel-6; 1.7 m height; Patura, Laudenbach, Germany), covered with opaque weatherproofing tarpaulin, which was fixed tightly to the metal of the panels to prevent rustling (Figure 1). The race narrowed from 2.6 m width at the barn opening to a single-file race of 0.9 m width in 2.5 m distance to the opening and ran straight to the corralled pasture, thus screening cows of NonH-R from visual distractions. It separated the exits of NonH and HABIT and prevented visual contact between the groups outside of the barn.

Behind each opening, there was a grid for dropping manure collected by automatic scrapers. Unprepared, those openings would be unsuitable for evacuating cattle, since the grid structure was wide enough for cow claws to slip through. Thus, the grids were covered by wooden boards with struts gripping the grid structure. The same rubber mats as those used as flooring in the cubicle alleys were tacked to the wooden boards to provide cows with familiar flooring at the opening and to prevent slipping.

The setup took place in the afternoon before evacuation. The fire brigade arrived at 8:00 p.m., one hour after sunset, with milking ending at around 5:00 p.m. They advanced with sirens, blue lights, and full gear, parking their vehicles realistically close to both sides of the barn. Sirens were shut down upon arrival, but blue lights were kept going until all cows were on pasture.

Although power cuts are common in barn fires, the low lighting inside the barn was not turned off to respect the safety of the herding personnel and to allow video footage of the evacuation. Outside, the fire brigade put up spotlights to illuminate the area in front of the barn as well as the pasture to which the cows were to be evacuated. The spotlights were positioned orthogonally to the escape routes in order to avoid blinding the animals.

Each group of cows was herded by one employee of the farm, known to the animals, and one firefighter, unknown to the animals. They herded the animals by moving towards them, waving arms and calling out but without touching the animals, as instructed, after additional farm personnel opened the lashing straps and moved aside.

Cows of all three groups were mixed and left on pasture for about 30 min before farm personnel herded them back to the barn and separating them again.

### 2.4. Measures and Data Collection

To evaluate the success of evacuation, we focused on the required time for cattle to leave the barn. However, assessment of stress reactions in the cows was of interest as well, since highly stressed cattle would be more likely to seek refuge in familiar surroundings and to resist being herded outside. Furthermore, panicking animals present a danger especially to bystanders or untrained handlers, including the fire brigade.

The time needed for evacuation was analysed by recording video footage. In use was a Dual-Sensor Camera (AXIS P3715-PLVE Network Camera; Axis Communications GmbH, Ismaning, Germany), which was fixed to the ceiling of the barn, aiming the lenses to both sides, capturing all three exits. In addition, camcorders were set up outside the barn. Two drones (DJI Mini 2; Da-Jiang Innovations Science and Technology CO, Shenzhen, Guangdong) were also in use, capturing the area outside the barn and the behaviour of cows on pasture. In the same manner, the habituation of HABIT to pasture prior to evacuation was recorded.

For evaluating physiological stress reactions in the cows, faecal samples were collected and FCMs were quantified. Sampling took place with cows in headlocks at the feeding fence after milking, 10 h ± 15 min after evacuation. In the same way, faecal samples of HABIT were taken at days 1, 2, 3, and 6 of habituation. Faecal samples of each cow were taken beforehand to determine their baseline FCM levels. Sampling for baseline took place in the morning after milking, when FCM concentration should reflect the serum cortisol levels of the cows while resting, 9 to 12 h beforehand. Cows of HABIT were sampled on the day before habituation, while cows of NonH and NonH-R were sampled in the morning of the day of evacuation. Faeces were taken manually from the rectum or directly during the process of defecating, but never from the ground, using disposable rectal examination gloves and filling faeces into sample tubes (Sample Container 70 × 24 mm, 17 mL Volume; Süsse, Gudensberg, Germany). During sampling, the behaviour of the animals was checked for unease using a Score Sheet with defined abort criteria. Samples were preserved at −20 °C in a deep freezer at the farm, directly after collecting and labelling them. They were then transported in a mobile freezer at −18 °C to the laboratory of the Chair of Animal Welfare, Ethology, Animal Hygiene and Animal Husbandry, Department of Veterinary Sciences, Faculty of Veterinary Medicine, LMU Munich, for extraction by dispensing 0.5 g faeces in 5 mL of 80% methanol and draining the supernatant after vortexing for 30 min and centrifuging at 2500× *g* for 15 min [33,39,40]. FCMs were then quantified using an 11-oxoaetiocholanolone enzyme immunoassay (EIA measuring 11,17-dioxoandrostanes). Details of the EIA are described elsewhere [41,42]. FCM concentrations are expressed in nanograms per gram of fresh faeces. The sensitivity of the EIA is 2.2 ng/g. Intra-assay and inter-assay coefficients of variations were below 10% and 12%, respectively.

The milk yield of cows in all groups was recorded twice a day in the parlour using flow meters. The combined milk yield in litres was corrected in accordance with German animal products law with the factor 1.03 to state the milk yield in kilograms.

### 2.5. Statistical Analysis

Statistical analysis was performed using IBM SPSS Statistics Version 26 and the open software R. Boxplots were produced for the data on the required time for evacuation, the milk yield, and the FCMs of the three different groups, stratifying for lactation if necessary. Data on milk yield and FCM were assessed visually for normality using these boxplots and Q-Q-Plots. As most data were not normally distributed, all analyses were conducted using non-parametric statistical tests. Data were analysed for differences between the three groups, using the Kruskal–Wallis test with subsequent Mann–Whitney U-tests for pairwise comparison using the Bonferroni correction. Baseline FCM concentrations were compared to the concentrations after evacuation using a paired Wilcoxon test. Relative differences (in %) in the FCM concentrations were compared using Kruskal–Wallis-tests. Differences in milk yield prior to evacuation (mean of 14 days prior to evacuation) and the individual milk yields in the seven days post-evacuation were analysed using Friedman’s test, a non-parametric test for paired data. The difference (in litres) between the mean daily milk yield of the three days post-evacuation and the 14 days prior to evacuation were compared between the three groups using Kruskal–Wallis tests with subsequent Mann–Whitney U-tests for pairwise comparison using the Bonferroni correction. The relationships between the change in FCM concentration and the change in milk yield before and after evacuation were analysed using Pearson correlation. All *p*-values less than 0.05 were considered statistically significant.

## 3. Results

### 3.1. Time Needed for Evacuation

Video footage of evacuation was analysed to capture the time needed for the first and last cow of HABIT, NonH, and NonH-R to leave the barn and to walk onto the corralled pasture after opening the lashing strips. There was a delay of only 14 s between the first cow and the last cow of HABIT to leave the barn. For cows in NonH, this delay was 38 s, and for cows in NonH-R, it was 73 s (Figure 2). This time period was statistically significantly shorter for HABIT than for the other two groups (HABIT vs. NonH: p_adj_ = 0.0001; HABIT vs. NonH-R: p_adj_ = 0.0002; NonH vs. NonH-R: p_adj_ = 0.1400). The mean time interval between individual cows leaving the barn was 0.67 *±* 0.56 s for HABIT, 1.62 *±* 1.21 s for NonH, and 3.19 *±* 1.61 s for NonH-R (*p* < 0.0001, with statistically significant differences between all groups: HABIT vs. NonH: p_adj_ = 0.0116; HABIT vs. NonH-R: p_adj_ < 0.0001; NonH vs. NonH-R: p_adj_ = 0.0072).

The response time from opening the lashing strips to the first cow leaving the barn was greatest for NonH-R with 87 s, followed by HABIT with 22 s and NonH with only 6 s (Table 1). The time it took for each animal from opening the lashing strips to leave the barn was statistically significantly different between the three groups (*p* < 0.0001). There was no statistically significant difference in this time duration between the cows in the HABIT and NonH group (p_adj_ = 1.0), but the times of cows in the NonH-R differed from the ones of the other two groups (p_adj_ < 0.0001 each).

Describing observations, cows of HABIT appeared to move as bulk, walking straight to the pasture (Figure 3). This is also reflected in the shortest distances between the cows during evacuation. Cows of NonH moved quicker; some tried to turn around and some showed standing bouts (Figure 4). Cows of NonH-R moved in single file through the race.

### 3.2. Faecal Cortisol Metabolites (FCMs)

#### 3.2.1. FCM Baseline

Baseline FCM concentrations in HABIT (median: 18.7 ng/g; min/max: 5.9/50.2 ng/g), NonH (median: 15.2 ng/g; min/max: 9.7/32.7 ng/g) and NonH-R (median: 20.5 ng/g; min/max: 5.0/61.6 ng/g) did not differ significantly (*p* = 0.2389; Figure 5). The median value of all cows was 17.4 ng/g. There was no significant difference in baseline values between Simmental (median: 16.8 ng/g; min/max: 6.6/61.6 ng/g; *n* = 30), Brown Swiss (median: 15.4 ng/g; min/max: 5.0/37.7 ng/g; *n* = 23), beef hybrids (median: 17.9 ng/g; min/max: 5.9/39.0 ng/g; *n* = 9), and Red Holstein (median: 25.9 ng/g; min/max: 14.2/50.2 ng/g; *n* = 6) (*p* = 0.3261). Regarding parity, there was no statistically significant (*p* = 0.5316) difference in baseline values between uniparous cows (median: 16.8 ng/g; min/max: 5.9/37.9 ng/g; *n*= 16) and multiparous cows (median: 17.4 ng/g; min/max: 5.0/61.6 ng/g; *n*= 53).

#### 3.2.2. FCMs during Habituation

During the habituation period of HABIT, four faecal samples were taken 10 h ± 15 min after herding the group of cows to pasture in the morning. The relative difference of FCM concentrations in relation to the baseline differed statistically significantly between the four sampling days (*p* < 0.0001), whereby the values of day one (median: 161%; min/max: −38/828%), day two (median: 115%; min/max: −35/1345%), and day 6 (median: 66%; min/max: −76/263%) did not statistically significantly differ from each other. However, the values of day 3 of habituation were the lowest (median: −28%; min/max: −74/107%) and statistically significantly differed from all the other days (day 3 vs. day 1: *p* < 0.0001; day 3 vs. day 2: *p* < 0.0001; day 3 vs. day 6: *p* = 0.0024) (Figure 6).

Remarkable was the behaviour of one particular Brown Swiss cow (9.45 years of age, low milk yield). The cow could not be persuaded to leave the barn, neither during habituation nor during evacuation. She did not seem stressed by being isolated, although FCM concentration did rise in a similar way as the rest of the group (baseline: 14.41 ng/g; H1: +394%; H2: +260%; H3: +17%; H6: +70%; evacuation: +126%).

#### 3.2.3. FCMs during Evacuation

The fire brigade arrived shortly after 8:00 p.m. on the evening of evacuation. At around 6:00 a.m., 10 h ± 15 min after evacuation, faecal samples were taken from all cows. The differences in the FCM concentrations after evacuation in relation to the baseline were statistically significant for two of the three groups (HABIT: *p* = 0.0003; NonH: *p* = 0.0130; NonH-R: *p* = 0.560) (Figure 7). The relative increase in FCM concentration was not statistically significantly different between the three groups (HABIT: median: 112%; min/max: −49/441%; NonH: median: 22%; min/max: −59/306%; NonH-R: median: −3%; min/max: −67/455%; *p* = 0.1736), with large variations between individual cows. There were no stattistically significant differences in the relative increase in FCM concentration between breeds (*p* = 0.4308) or between uniparous and multiparous cows (*p* = 0.1145) and no correlation with milk yield (*p* = 0.1907).

The change in FCM after evacuation was not correlated with daily milk yield reduction (Pearson correlation r = 0.0440, *p* = 0.07195).

### 3.3. Changes in Daily Milk Yield

The milk yield of all cows was recorded twice a day at the parlour with flow meters. There was a statistically significant difference in the mean milk yield 14 days prior to evacuation between the three groups (*p* < 0.0001), whereby the mean milk yield of HABIT was significantly lower than that of NonH (p_adj_ = 0.0001) and lower than that measured in NonH-R (p_adj_ < 0.0001). The mean milk yield of NonH differed not statistically significantly from that of NonH-R (p_adj_ = 0.7351). This was also the case for the mean milk yield of the three days following the evacuation (*p* < 0.0001; HABIT vs. NonH: p_adj_ < 0.0001; HABIT vs. NonH-R: p_adj_ < 0.0001; NonH vs. NonH-R: *p* = 0.3800).

No statistical significant difference was observed in the change in mean daily milk yield between the 14 days prior to evacuation compared to the three days post-evacuation between the three groups (*p* = 0.0707).

Comparing the mean daily milk yield during the 14 days prior to evacuation to the mean of daily milk yield during the first 3 days past evacuation, a noticeable drop was observed in HABIT (Wilcoxon paired test *p* = 0.0002), but not in NonH (Wilcoxon paired test *p* = 0.1564) nor in NonH-R (Wilcoxon paired test *p* = 0.6702) (Table 2).

Investigating the course of the milk yield within the three groups, comparing the mean milk yield 14 days prior to evacuation with the seven individual days after the evacuation did not show any statistically significant differences in the milk yield within the groups NonH (Friedman’s test *p* = 0.1849) and NonH-R (Friedman’s test *p* = 0.1054). Only in the HABIT group, there was a statistically significant difference (Friedman’s test *p* < 0.0001), with day 2 after evacuation showing the lowest milk yield (−2.75 ± 3.61 kg compared to the 14 days prior to evacuation).

In HABIT, a statistically significant decrease in daily milk yield was also observable, comparing the mean during the 14 days prior to the habituation period to the mean of HABIT during the first 3 days of habituation (−1.65 ± 1.42 kg; Wilcoxon-paired test *p* < 0.0001).

## 4. Discussion

The results of this first explorative study, concerning the evacuation of cattle, indicate the effectiveness of exits, which are adjusted to the sensory physiology of the animals, to respect animal welfare and to prevent losses caused by fire or natural disaster.

The range of FCM values differed strongly between individual animals, as was also shown in the literature [41,43]. With median baseline of all cows at 17.4 ng/g, our results fit the findings of previous studies, falling, for example, in between the results of Rouha-Mülleder et al. with median baseline of 23.4 ng/g [43] and of Ebinghaus et al. with 11.0 ng/g [36].

Since analytical methods as well as sampling, storing, and transport influence the results of FCM quantification, a direct comparison of values between different studies might not be informative. In addition, the influence of health status as well as milk yield on the FCM concentration is not yet fully explored. Rouha-Mülleder et al. found a correlation between decubitus at tarsal joints and higher FCM concentrations [43]. Ivemeyer et al. found a correlation between lower baseline FCM values and the rate of self-curing udder disorders, defined by staying below 100,000 somatic cell count over three months following a somatic cell count of over 200,000 [44]. Bertulat et al. found differences in baseline FCM concentrations among low-, medium-, and high-yielding groups of cows, with lower values for the latter [45]. On the other hand, Pesenhofer et al. found no differences in FCM concentration between cows without lameness and cows with scores indicating slight or marked lameness [35]. To account for such individual differences, it has been suggested that each animal acts as its own control [38,40]. Therefore, the individual FCM rise (in relation to its baseline) following the stressful event was used to assess the physiological reaction of each cow to the event.

Sampling once after a stressful event, FCM values can depict the general level of stress the animals were affected with over a stretch of time by interpretation of the rise in concentrations, but without reliably depicting the peak, which would require repeated sampling. Pesenhofer et al. measured the peak of FCM concentration 9 h after a stressful event [35]. Heinrich et al. noted a plateau in FCM concentration after 9 h, rising only slightly until 11 h after the occurring stress [37]. Palme et al. described that a peak in FCM followed a peak in plasma cortisol concentration 10 h later [41], while Möstl et al. found that a peak in FCMs in cattle occurred 12 h after transportation [34]. To achieve the main objective of this study, the comparison of evacuation behaviour between cows in HABIT, NonH, and NonH-R, it was not mandatory to pinpoint safely the peak of FCM rise after evacuation, but to compare the rise in FCMs after the same amount of time between the groups. Repeated measurements might have revealed more details in FCM excretion.

For cows in HABIT, being younger on average than cows in NonH and NonH-R, less time passed on average since they were on pasture at the farm site where all cows were kept as young cattle. This should not have had an interfering effect on the rise in FCMs after evacuation, for there was no significant difference between uniparous and multiparous cows. However, cows in HABIT being younger could have influenced the quickness of habituation. The return to baseline values in FCM concentrations already on day three of the habituation period implies a swift and successful familiarisation of cows in HABIT.

The rise in FCM levels after evacuation in this study was comparably moderate. It was less than the rise in FCMs during heat stress exposure, as reported by Veissier et al. [46]. For HABIT, the rise in FCMs was comparable to the change in FCMs that Kuhlberg et al. noticed in heifers between one week prepartum and two weeks postpartum [47]. Our results indicate cows sufficiently accepting the designed openings, allowing for swift rescue. This is contrary to our expectations, resulting from field reports of firefighters and affected farmers, emphasising the difficulty in moving unhabituated cows out of the barn. One influencing factor may be that cows on the educational and research farm were more used to foreign persons and procedures than cows on commercial dairy farms.

In conclusion, possible differences in the amount of stress experienced in the moments of leaving the barn, resulting from the different designs of evacuation between the groups, were not high enough to significantly affect the whole level of stress on the evening of evacuation. More likely, the FCM concentrations of the samples taken the following morning depicted imposing stress by mixing groups on pasture and by herding cows back to the barn, as well. While highlighting intriguing directions of future research, this study can neither differentiate nor quantify the influences of the factors investigated.

Reactions in the hypothalamic–pituitary–adrenal axis to a stressful event can be evaluated by FCMs. Additionally, reactions in the sympathoadrenal axis of the autonomic nervous system, regulating variations in the heart rate, can be depicted to assess stress. Concerning follow-up studies, the option of using sensors recording intervals between two following R-waves (RR-Intervals) to register the heart rate variability (HRV) at various defined points of time during evacuation should be considered to improve differentiation [48].

Surprisingly, the reduction in milk yield after evacuation was only statistically significant in HABIT and was not dependent on the lactation number of the cows. Additional stress caused by the sudden halt of access to pasture in the morning might have influenced this. Our results concerning milk reduction are comparable with the findings of Pesenhofer et al., describing a decline in the median of daily milk yield by 0.6 L on the day after claw trimming in comparison to the median of the 7 days prior to claw-trimming [35]. Gräff found a mean reduction of 0.55 L 10 days past a simulated power cut of an automatic milking system in comparison to the 10 days prior [49]. However, Broucek et al. noticed a reduction of 23.3% in daily milk yield on the first day after transferring a herd of Holstein cows from stanchion-stall housing to a new facility with freestall housing [50].

Concerning the design of the openings, the access to cubicles seemed to have an impact on the response time between opening the lashing straps and the first cow leaving the barn. For NonH, this was only 6 s, being herded through an opening at the end of the feed passage, without access to cubicles. For NonH-R, this was 87 s, with cows firstly evading the pressure built by the herding personnel by retreating in cubicles instead of choosing the opening through the race upon pasture. For HABIT, despite having access to cubicles, it took only 22 s until the first cow left the barn. This response time, besides differences in access to cubicles, might have also been influenced by hard-to-standardise herding pressure of the firefighters and the farm personnel, as well as random positioning of cows in front of the exit at the moment of opening the lashing straps.

After the first cow of NonH-R went into the race, the other cows followed in a mostly steady line. Since it was a single-file race, the interval between individual cows leaving the barn was considerably longer than with NonH or HABIT, for which the openings were wide enough for two cows to fit through simultaneously. The swiftness of cows in NonH to leave the barn might have been influenced by being herded in the broader feed passage in addition to having an opening that was slightly wider than openings for NonH-R and HABIT. However, the delay between the first and last cow leaving the barn was still significantly longer with NonH than for HABIT.

Neither the hypothetical advantage of the race blocking the sight of cows to the vehicles, personnel, lights, and other potential stress inducing distractions, nor the potential advantage of the race to prevent cows from turning around and retreating back into the familiar barn have proven to be effective or rather necessary with the presented experimental setting. Although a few cows in NonH turned around, trying to get back into the barn and escape the unknown impressions outside, the herding personnel were able to dissuade them easily. This was certainly supported by the positive effect of the design of the escape routes, which were adapted to fit the sensory physiology of cattle, with focus on lighting and surface, following the guidelines of the extensive work of Temple Grandin [30,31,32]. We were surprised nonetheless by the apparent effectiveness of these slight adaptations. While the positioning of the lights at night by the fire brigade is adaptable, bright sunshine in the daytime is not, and can be glaring for cattle, as well. If possible, an evacuation route with a shaded opening, averting the sun, should be considered.

With cows in HABIT, turning around outside the barn was not observed. They seemingly were able to transfer their habituation of the route to pasture, which they previously learned was safe in calm circumstances, to a stressful evacuation with the fire brigade present. They left the barn readily in bulk, with the lowest time needed between the first and last cow leaving the barn. Outside, they seemed rather relaxed, with some cows watching the fire brigade while standing on the concrete area between barn and pasture. They were not trying to get back to the barn and were more laid-back reacting to the pressure of the herding personnel than cows in NonH. This observed calmness was not depicted by a significant difference in FCMs between HABIT and NonH or NonH-R. An evacuation late at night might have been more conclusive, with cows being even more inactive in the middle of the resting period instead of right after sunset.

Because, to our knowledge, there are no comparable studies to predict animal behaviour, it was necessary to start the experimental design with a low array of stressors. This was carried out in accordance with the rule of refinement, formulated by Russel and Burch in 1959 [51], and consequential to being mindful of experimental safety, especially towards the herding participants of the fire brigade with less experience concerning animal behaviour. However, the moderate change in FCM levels in all groups compared to baseline implies that a more realistic experimental design might be necessary and ethically acceptable. To highlight further effects of different designs of openings on the success of evacuation, additional stressors might be needed, simulating a more realistic emergency scenario and ensuring the applicability of the results. Subsequent studies should explore, for example, the usage of hot smoking installations inside the barn or locally controlled fires, improving the realism of the study design concerning sight, smell, and temperature.

## 5. Conclusions

With the correct preparation, the evacuation of a herd of lactating cows seems to be feasible, even if they were not previously habituated to leaving the barn. However, cows that were habituated to the exit beforehand left more rapidly. This study encourages farmers to think about possible means of evacuating cattle in case of fire or natural disaster and to give instructions regarding the design of egress. Further research is necessary for better understanding the factors influencing the success of an evacuation.

## Figures and Tables

**Figure 1 animals-12-01344-f001:**
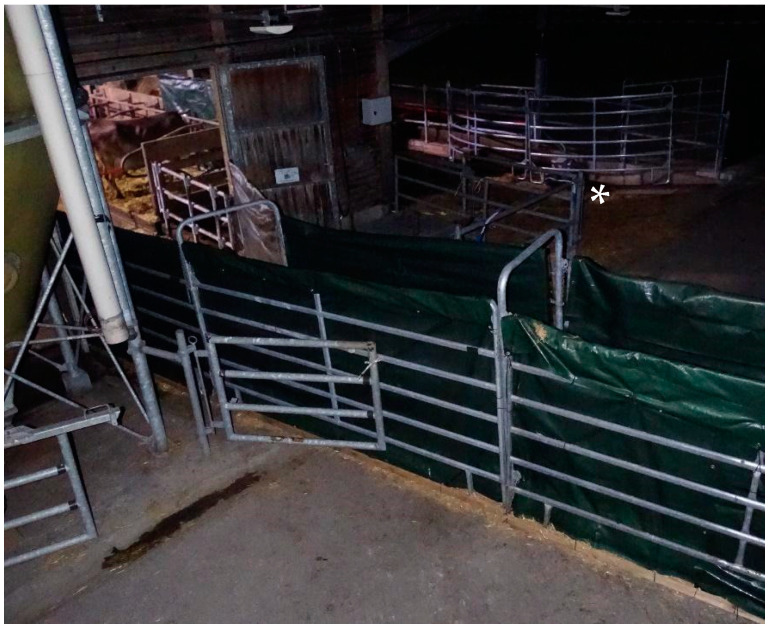
Race for NonH-R, covered with opaque weatherproofing tarpaulin. In the background, the exit of NonH is visible (*). The exit of HABIT would be to the left.

**Figure 2 animals-12-01344-f002:**
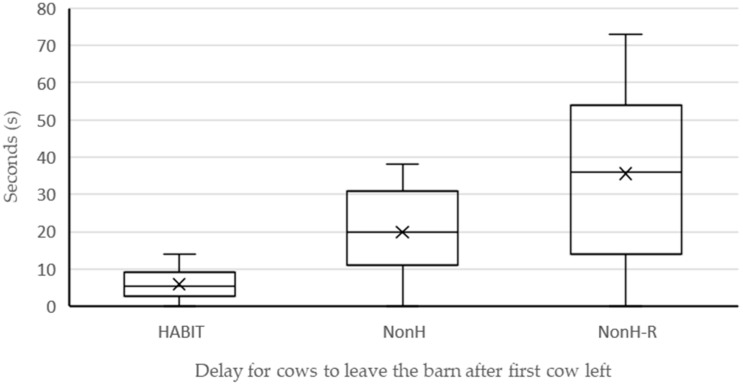
Boxplot graphs (bold line: median; cross: mean value; boxes: first and third quartile; whiskers: 5th and 95th percentiles) of time passed, depicted in seconds, between first cow and each following cow leaving the barn per group.

**Figure 3 animals-12-01344-f003:**
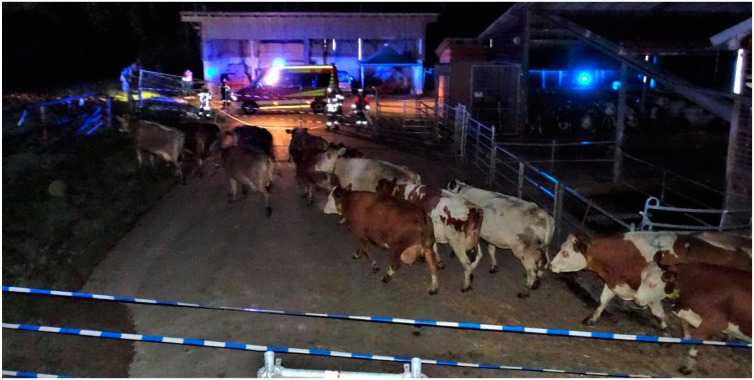
Cows of HABIT leaving the barn with fire engine next to the route to pasture.

**Figure 4 animals-12-01344-f004:**
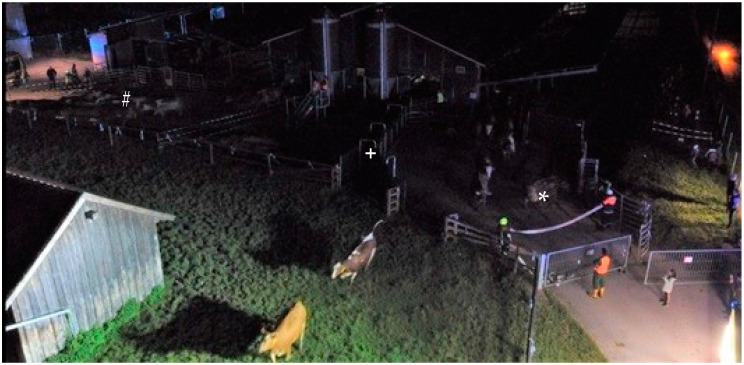
Cows of NonH (r.) leaving the barn with a Brown Swiss (*) turning around on the right side. Cows of HABIT leaving the barn in a bulk in the far left (#). Race of NonH-R in the middle with no cow leaving the barn yet (+).

**Figure 5 animals-12-01344-f005:**
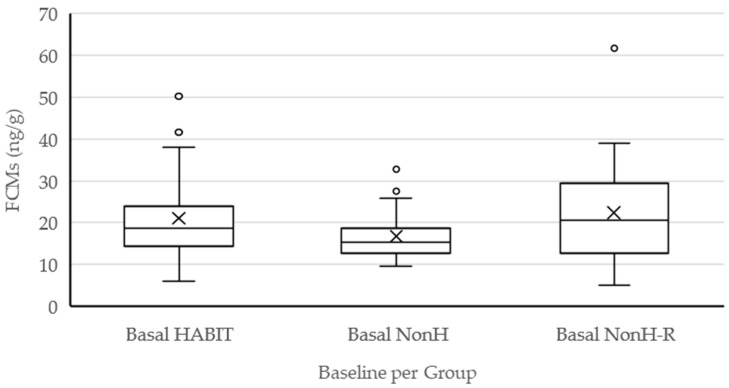
Boxplot graphs (bold line: median; cross: mean value; boxes: first and third quartile; whiskers: 5th and 95th percentiles; circles: outlier values) of baseline FCM concentrations (ng/g) per group.

**Figure 6 animals-12-01344-f006:**
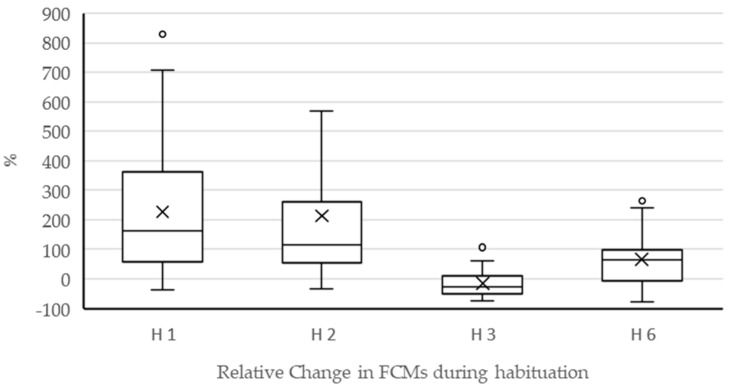
Boxplot graphs (bold line: median; cross: mean value; boxes: first and third quartile; whisk-ers: 5th and 95th percentiles; circles: outlier values) of individual differences in FCM concentrations (%) between baseline samples and samples on day one (H1), day two (H2) with one outlier of 1345% not depicted, day three (H3), and day six (H6) of habituating HABIT to pasture.

**Figure 7 animals-12-01344-f007:**
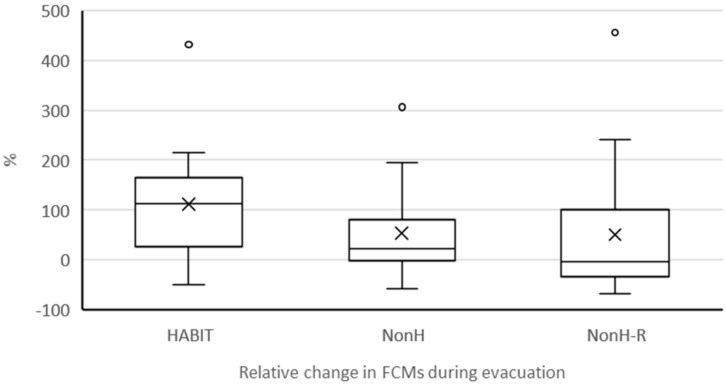
Boxplot graphs (bold line: median; cross: mean value; boxes: first and third quartile; whisk-ers: 5th and 95th percentiles; circles: outlier values) of individual differences in FCM concentrations (%) between baseline samples and samples after evacuation per group.

**Table 1 animals-12-01344-t001:** Time needed (mm:ss) for the first and last cow per group to leave the barn and to arrive at the corralled pasture.

	HABIT		NonH		NonH-R	
	Barn	Pasture	Barn	Pasture	Barn	Pasture
First Cow	00:22	00:32	00:06	00:12	01:27	01:40
Last Cow	00:36	01:15	00:44	00:52	02:40	02:51

**Table 2 animals-12-01344-t002:** Differences between the mean daily milk yield during the 14 days prior to evacuation and the mean daily milk yield (kg/d) during the three days past evacuation per group.

	Mean 14 d Prior E	Mean 3 d Past E	Difference
HABIT	27.77 ± 4.13	26.02 ± 4.14	−1.75 ± 1.67
NonH	39.64 ± 6.00	39.04 ± 5.88	−0.60 ± 1.91
NonH-R	41.04 ± 3.77	40.9 ± 4.95	−0.14 ± 2.46

## Data Availability

The data presented in this study are available on request. Please contact florian.diel@hswt.de.

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
