# Peer review of "Exploring the Evacuation of Dairy Cattle at Night in Collaboration with the Fire Brigade: How to Prepare Openings for Swift Rescue in Case of Barn Fire"

_animals, 2022, doi:10.3390/ani12111344_

Round 1

Reviewer 1 Report

This study is highly evaluated as the first evacuation study  of cattle.

The conclusion that cows habituated during daytime more rapidly, seeming being able to transfer this learning to a stressful evacuation at night is a little overestimated. Cows might be still active at 8pm, one hour after sunset, when this study started. Experiment must be work out during inactive time of cows around  2am, if authors wish to transfer this habituation learning to evacuation at night.

Author Response

Thank you very much for your review and the constructive criticism.

I adapted the wording of our findings in this regard, leaving out the preparedness for an evacuation in the middle of the night. Also, I added thoughts, discussing possible following study designs:

Lines 531-533: "An evacuation late at night might have been more conclusive, with cows being even more inactive in the middle of the resting period instead of right after sunset."

Nevertheless, Cows seemingly were able to transfer their training from daytime to the different lighting at night. Certainly further studies are necessary.

Kind regards,
Florian Diel

Reviewer 2 Report

The paper originally reports a topic that has not yet been the subject of in-depth studies. Therefore, the initiative conducted by the authors in collaboration with the firefighters is appreciable. 
Only a minor revision is required, with some comments pinned in the attached file.

Author Response

Thank you very much for your help and review. I revised the paper accordingly.

Concerning your first comment at page 12, I feel the "it" to be necessary:

“To achieve the main objective of this study,..., it was not mandatory to pinpoint safely the peak of FCM rise after evacuation,...”

I hope not to have slipped concerning this.

Kind regards,

Florian Diel

Reviewer 3 Report

Regardless of the general formulation of the research / study goal, the article should also include the sentences presenting the cognitive (scientific) goal of the research and the utilitarian (useful) goals formulated by the authors in the research? The overview of the knowledge in the Introduction chapter could be summarized in the form of a sentence describing the research problem. I suggest that you write: "The research problem is ...". The research problem can be related to the indication of a gap in the current state of knowledge, which inspired the research. In general, the considerations related to the gap in the current state of knowledge were undertaken by the Authors, but it should be clearly called a gap. Considerations related to the practical aspects of research were also presented by the Authors at the end of the Introduction, but they should be called the practical goal of the study.

The authors considered the case of the evacuation of dairy cattle in connection with a fire at night. In practice, there are also fires of livestock facilities outside the time of night (during the day)? In the event of a fire, would the procedures for animals and human behavior be the same as at night? Perhaps it would be worth mentioning this in the article as well.

I would like to ask if the authors tried to simulate the smell associated with the fire, i.e. smoke, in the experiment. Such smoke can be produced artificially and given the right direction of flow in the building. In this way, it would be possible to get closer to the real conditions characterizing the fire, taking into account the controlled (and safe in the experiment) smoke phenomenon that accompanies the fire. A similar point applies to the lighting effects that are associated with a fire. Did the authors think about the possibility of getting closer to the real conditions prevailing during the fire by means of lighting effects and their modulation in the barn or near the barn? In my opinion, artificial smoke and modulated light at night (because this time was chosen for the experiment) would allow for a more detailed understanding of the animals' reactions, including their behaviour during evacuation. It would also allow for a more accurate diagnosis of reactions in the organisms of animals related to the secretion of cortisol, but also changes in the daily milk yield.

The fire phenomenon is also accompanied by an increase in temperature. So I would like to ask if the authors tried to influence the temperature changes during the experiment? If such attempts have not been made, it can be written in the article that this is a suggestion for future research that can be continued, also with other groups of livestock.

Author Response

Thank you very much for your review and the constructive criticism. I tried to adapt the paper accordingly. Comments mark the relevant passages in the text. Changes are as follows.

Lines 44-45: "The research problem, concerning the evacuation of cattle in case of barn fire, is the apparent gap in the current state of knowledge."

Lines 116-118: "In an effort to reduce the amount of farm animals perishing in barn fires, this pilot study was undertaken. The practical goal of the study is to explore possible designs of egress and strategies evacuating cattle. The supporting, more scientific goal is to explore stress responses in cattle while evacuation.

Line 96: A path orderly lighted by the fire brigade in the night, or bright sunshine at daytime might be glaring and blinding for cattle. 

Lines 518-520: While the positioning of the lights at night by the fire brigade is adaptable, bright sunshine at daytime is not and can be glaring for cattle as well. If possible, an evacuation route with a shaded opening, averting the sun, might be considered. 

In further research, the possibility of a more realistic scenario should be the goal. In this first attempt, we tried to start deliberatly with a low array of stressors, as described at the end of discussion. The creation of safe, artifical smoke would be an option, but without the properties of "real" smoke in a barn fire, produced by burning plastics, silage, bedding etc. we felt that artificial smoke would hinder recordings of the experiment, affect experimental safety especially towards the herding personel and could not with certainty improve the realism of the study design by much. Nevertheless, the use of locally controlled fires or hot smoking installations is suggested for following research at the end of discussion. I adapted as follows:

Lines 544-545: "Subsequent studies should explore for example the usage of hot smoking installations inside the barn or locally controlled fires, improving the realism of the study design concerning sight, smell and temperature."

Kind regards,
Florian Diel